# Observing COVID-19 related behaviors in a high visitor use area of Arches National Park

**Zachary D. Miller**[1,2,3][◉]*, **Wayne Freimund**[2,3,4][◉], **Douglas Dalenberg**[5][◉], **Madison Vega**[1]

**1** Department of Environment and Society, Utah State University, Logan, Utah, United States of America,
**2** Institute of Outdoor Recreation and Tourism, Utah State University, Logan, Utah, United States of America,
**3** The Ecology Center, Utah State University, Logan, Utah, United States of America, **4** Department of
Environment and Society, Utah State University, Moab, Utah, United States of America, **5** Department of
Economics, University of Montana, Missoula, Montana, United States of America

◉ These authors contributed equally to this work.
* zachary.miller@usu.edu

## Abstract

### Introduction

Visitation to parks and protected areas is a common COVID-19 coping strategy promoted
by state and national public health officials and political leadership. Crowding and conges-
tions in parks has been a perennial problem and the ability to socially distance within them is
an unproven assumption. Is it possible to socially distance in a busy national park that has
been designed to concentrate use?

### Methodology/Principal findings

An observational study was conducted in July 2020 at the outside foyer of the Visitor Center
of Arches National Park. Motion sensor cameras were placed to record one-minute videos
when a person entered the field of view. Number of groups, group size, facial coverings and
encounters within 6 feet (1.83 meters) of other groups were recorded. Groups were smaller
on average than recorded in previous studies. Approximately 61% of the visitors wore
masks. Most groups (69%) were able to experience the visitor center with no intergroup
encounters. We model the probability of intergroup encounters and find as group size and
number of groups increases, the probability of encounters rises. With four groups present,
the probability of one or more encounters ranges from 19% to 40% for common group sizes,
while if eight groups are present, the probability of one or more encounters increases from
34% to 64% for common group sizes.

### Conclusions/Significance

Under conditions in which park visitors have the physical space to avoid close encounters
with other groups they are taking advantage of the opportunity. Visitors are minimizing
group size, wearing masks, and remaining socially distant. However, encounters increase
as the number or the size of the groups increases. In other areas of the parks this ability to
avoid encounters may not be as possible. We recommend that park managers continue to

doi.org/10.1371/journal.pone.0247315

of Technology, ISRAEL

**Data Availability Statement:** Some data cannot be
shared publicly due to confidentiality issues of
images associated with the observational studies.
However, the minimal data set required to replicate
the study results is uploaded to Utah State

University's official institutional data repository (Digital Commons at Utah State University) and publicly accessible via https://doi.org/10.26078/ymtv-2w67.

**Funding:** The author(s) received no specific funding for this work.

appeal for compliance with CDC guidelines, especially the wearing of masks and encouraging visitors to split up into small groups when visiting.

## Introduction

Early 20[th] century national park advocates claimed that thousands of tired, nerve-shaken people would find that the wildness of national parks was a necessity [1]. Although they were wrong about many things [2], their foresight about masses of people heading to national parks for a sense of restoration was mostly correct. What they failed to anticipate in the early 1900s was that it would not be thousands, but rather hundreds of millions of people that would retreat to the national parks of the United States (US) during one of the most challenging times in early 21[st] century: the COVID-19 pandemic.

National parks have long been reservoirs for human health and well-being [3], and the COVID-19 pandemic reaffirmed that visitor experiences in parks and protected areas provide these essential services. After the initial closure of national parks in the US early in the COVID-19 pandemic [4], visitor use to national parks almost immediately rebounded. In fact, national parks in the US are recording visitation levels that exceed pre-pandemic levels [5], and some re-opening national parks saw such high demand that they immediately shut down due to traffic congestion [6]. Although the data indicate fulfillment of the idea that masses of nerve-shaken people would go to national parks to be restored, the high concentration of visitors in national parks during a global pandemic presents an unknown risk to the very health and well-being people seek. This presents a challenge for the managers of parks and protected areas like national parks.

In the State of Utah, the home of five national parks, 10.7 million visitors accounted for 1.2 billion US dollars in visitor spending just within gateway communities in 2019 [7]. To provide this scale of visitor access to parks and protected areas in a way that reduces risks related to COVID-19, managers need to be able to make science-informed decisions about visitor use. The fundamental research question in this manuscript is: *What explains visitors' ability to socially distance in a high visitor use area of a national park*? Observational data recorded visitor behaviors, and descriptive statistics and modeling explored visitor COVID-19 risk behaviors. The findings from this work can help managers of parks and protected areas better manage COVID-19-related risks while seeing near-recording breaking levels of visitor use in national parks.

### Risk-reducing behaviors during the COVID-19 pandemic

Arches National Park (ARCH) is in southeastern Utah, United States. The park is 4 miles (6 km) north of the town of Moab, Utah. More than 2,000 natural sandstone arches are located in the 76,679 acre (310,308,903 square meter) park. The park is primarily accessed by a single road and the vast majority of approximately 1.6 million annual visitors congregate at the visitor center and within a few popular areas that are accessible via 1 to 3 mile (1.6 to 4.8 km) trails from the road.

On March 27, 2020, the Utah Governor initiated a Stay Safe, Stay Home rule for the state of Utah that facilitated county health departments to place restrictions on camping to non-county residents, overnight lodging, in-restaurant dining and other essential services. These precautions along with concern for the health of park staff led to a closing of ARCH for a portion of March, the entirety of April, and most of May of 2020. During that time, outdoor

recreation was emerging as a "safe" alternative to in-house quarantine and was being encouraged by political leaders such as the state's governor and many public health officials. By the time ARCH reopened on May 29[th], demand was high enough that park managers were forced to move into a phased entry approach in which visitors had to wait for up to three hours for admission. During the three days of May that the park was open, nearly 14,000 recreational visits occurred [8]. During the month of June, the park had returned to use levels of over 163,000 visits and by July it recorded approximately 194,000 visits which is comparable to the July 2019 figure of 208,993 visits.

Use levels and potential for crowding have been an issue of concern at ARCH for many years [9, 10]. Given the popularity of the park, this means that hundreds of thousands of visitors monthly are entering a system in which they are funneled via the parking, facility, and trail infrastructure into close proximity with one another. Within ARCH, they are likely to encounter many of other visitors as they walk up to and view the arch formations.

Anticipating the potential for spread of COVID-19, the National Park Service (NPS) is striving to mitigate risk for both the visitors and their staff. For example, as many services as possible are being moved outside, hand sanitation is being provided throughout the park, and facilities are being cleaned at an accelerated rate. Personal responsibility guidelines are being communicated to visitors at many points including trailheads. Those messages comply with the Center for Disease Control guidance on travelling [11] a trailhead sign includes:

1. If you feel sick, please visit another day.

2. Practice social distancing. Maintain at least six feet distance between you and others.

3. Wear a face covering when social distancing cannot be maintained as mandated within Grand County [12].

4. Wash your hands often with soap and water for at least 20 seconds or use hand sanitizer.

5. Cover your mouth when you cough.

6. Avoid touching your eyes, nose and mouth.

The question posed in this study was whether it was physically and socially possible for visitors to comply with the guidance provided by the CDC and the NPS. Given that many parks are designed to minimize the human footprint on the natural resources, visitors are congregated by design in campgrounds, parking areas, amphitheaters, and visitor centers. Managers of ARCH were particularly concerned about places in the park where people were likely to wait for a service such as the restrooms or visitor centers. We chose to examine the social distancing behavior at the park Visitor Center.

## Materials and methods

### Site description

The Visitor Center at ARCH is located near the primary park entrance and is the first facility that visitors encounter. It is designed to accommodate a large and steady flow of visitors and provides approximately 140 parking spaces for vehicles and 10 for larger RVs or vehicles pulling trailers. Interpretive panels are permanently posted outside and an information center and souvenir store are in the building. Restroom facilities are accessed from outside. As a response to COVID-19, rangers were stationed outside to provide visitors with orientation information. A subset of materials from the bookstore was also provided outside. Thus, people who preferred to remain outside could meet most of their needs.

### Data collection

Data collection was completed using a Bushnell™ Trophy HD™ game camera. A single game camera was placed outside the visitor center in ARCH (see Fig 1). The game camera was placed out of reach of visitors. The game camera had a wide view of the plaza outside of the visitor center, and tape was placed over the lens of the camera to obscure any personally identifying features. The camera was set to capture one-minute-long videos any time motion was detected. Collection was approved through the Utah State University Institutional Review Board (IRB# 11273).

### Sampling details

Video was collected on July 12, 2020 and July 14, 2020. One full minute of video was used for the sample and the camera needed a brief period of time to reset and would trigger again when it detected motion. ARCH is set in the desert with ground temperatures exceeding 130 degrees Fahrenheit (54.4 degrees Celsius) regularly in July afternoons, and as a result the heaviest visitation occurs in the mornings (see Fig 2 showing entrances peaking between 9 and 11:59 AM). The extreme temperatures also impacted the camera operation by interfering with the motion detection, so the sample of video images was restricted to when the camera operated properly yielding images on July 12 from 7:17 AM to 11:10 AM and on July 14 from 7:18 AM to 7:58 AM, 10:04 AM to 10:54 AM, and 11:24 AM to 11:47 AM (see Fig 3). As shown in Fig 2, we have captured the busy morning period when encounters would seem to be most likely.

### Analysis

**Recording visitor behaviors.**   Videos were viewed by a university-trained researcher to record five behaviors of interest: group affiliation, group size, number of groups, number of people wearing masks, and intergroup interactions at a distance of <6 feet (1.83 meters). One

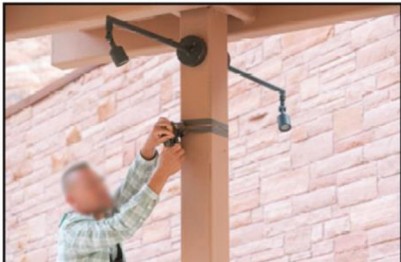

**Installing game camera**

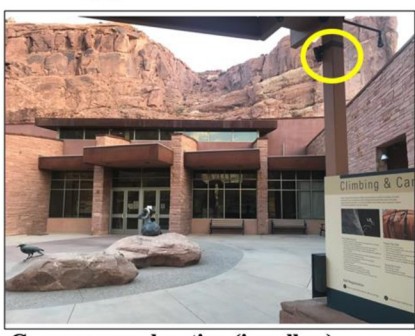

**Game camera location (in yellow)**

**View from game camera**

**Fig 1. Installation, location, and view from game camera.**

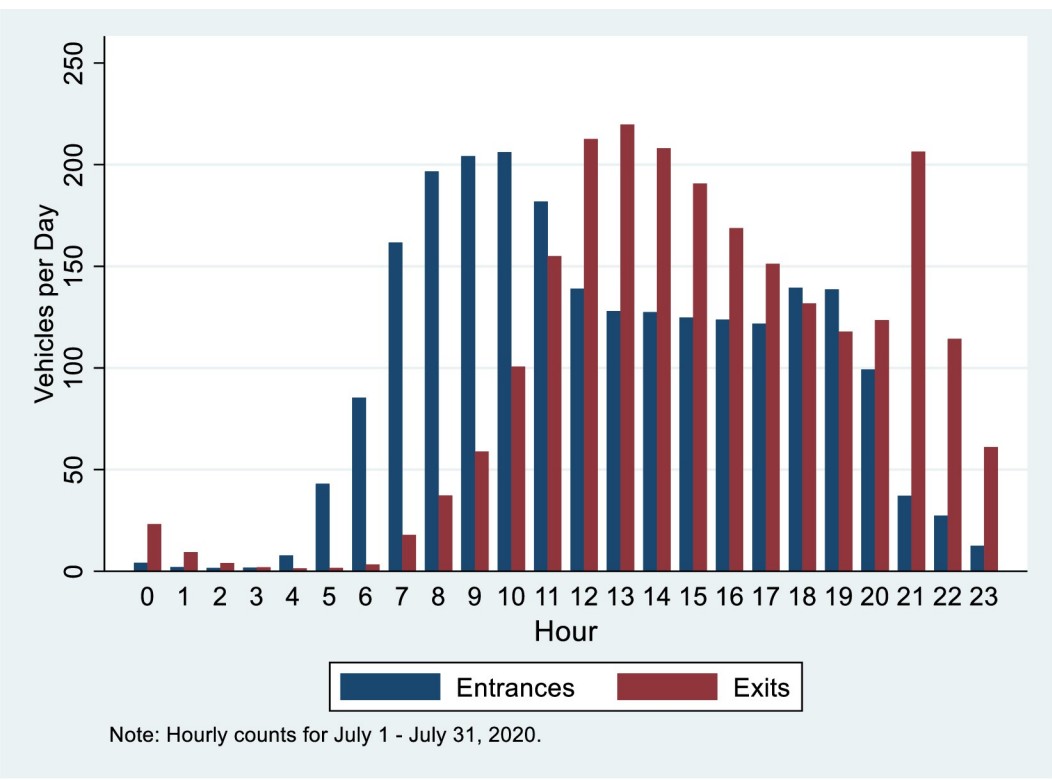

**Fig 2. Entrance and exit data for Arches National Park.**

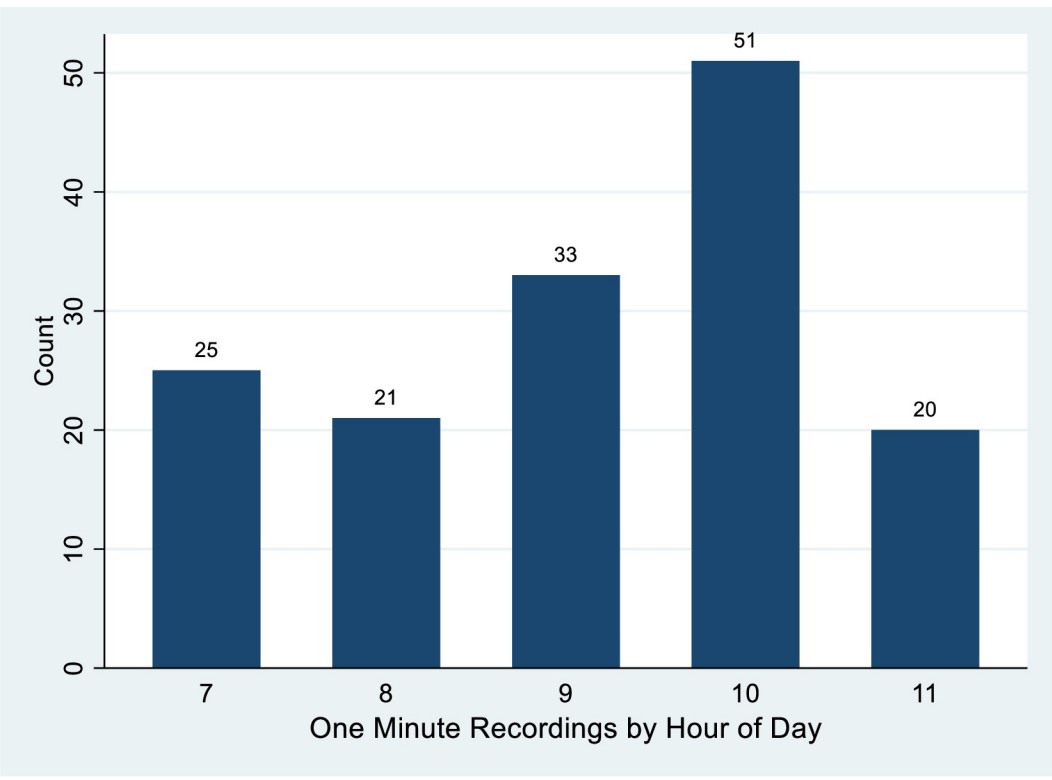

**Fig 3. Count of minutes observed by hour at Arches National Park visitor center on 7/12/2020 and 7/14/2020.**

person was responsible for the majority of the coding, and other members of the research team checked their coding in randomly selected clips to ensure reliability. Group affiliation was recorded using observable social cues to determine if people were affiliated in the same group or not. Examples of observable social cues include two or more people walking together within six feet of each other, at the same speed, and in the same general direction. Physical touch such as holding hands or passing items to others was also used to determined group affiliation. This data was used to calculate the total number of groups observed per minute, which represents a measure of visitor use density. Number of people was the number of unique individuals observed and was equal to the sum of the group sizes over the minute interval. Number of people wearing masks was the number of individuals that could clearly be seen wearing a face covering (i.e. cloth mask, disposable mask, bandana, etc.). If an individual was facing away from the camera for the entire recording or was seen with a face covering not worn properly over the nose and mouth for the entire recording, they would not be counted as wearing a face covering. The percentage of people wearing a face covering per group was calculated from this data. Intergroup interactions at a distance of <6 feet (1.83 meter) within the observed minute were determined based on estimated proximity between groups using reference points in the videos. For instance, we assumed adult height to be approximately 5–6 feet (1.52–1.83 meters) and used this to estimate interaction distances. If any individual in a group interacted with any individual from another group at a distance estimated to be <6 feet (1.83 meters), it was recorded as an interaction for the entire group. Other reference points, such as the length of the colored sidewalk portions, were measured on site. The use of reference points was necessary due to the 2-dimensional nature of the videos. There may, however, be an unknown amount of error related to the use of these reference points during observations. To counter some of this unknown error, the data recorder could classify the encounter as definite or probable; less than ten percent of the encounters fell into the probable category. We used the more conservative definition of an encounter and did not include probable encounters in the analysis.

**A count model for explaining increased encounters.** We hypothesized that the number of encounters in the observed video is associated with the group size, the number of groups, the number of people present, and the proportion of people in a group wearing masks. As the group size grows the chances of an encounter grow, similarly as the number of groups or people grows, holding the observed group's size constant, we expected more encounters. It is unclear whether mask wearing will lead to fewer encounters due to avoidance or more encounters due to a perception of improved safety. Since encounters is a count, we employed a count model. The estimation of the encounters model was performed with Stata v15.1.

## Results

### Observation statistics

Since the unit of observation is the behavior of a group in the one-minute period video, we have 780 observations of groups. The descriptive statistics for the sample are shown in Table 1. The distribution of encounters is shown in Fig 4. Sixty-nine percent of groups had zero encounters. On average there are 0.39 encounters by a group.

The average group size was 1.88 people and the distribution of groups shows that over half of the groups were single individuals as seen in Fig 5. This distribution of group size differs considerably from a recent study in which only 7 percent of the park visitors were in a group size of one [13].

The average number of groups observed potentially interacting with one another is 6.33 with a minimum of 1 group and a maximum of 11 groups during a one-minute interval (Table 1). The distribution of groups observed is shown in Fig 6.

**Table 1. Descriptive statistics of observations.**

|  | Median | Mean | Std. Dev. | Min. | Max. |
|---|---|---|---|---|---|
| Encounters[1] | 0.0 | 0.39 | 0.67 | 0 | 6 |
| Group Size | 1.0 | 1.88 | 1.19 | 1 | 8 |
| Groups[2] | 6.0 | 6.33 | 2.29 | 1 | 11 |
| People[3] | 12.0 | 11.93 | 5.28 | 1 | 26 |
| Percent Masked[4] | 100.0 | 60.91 | 46.17 | 0 | 100 |
| N | 780 |  |  |  |  |

[1] Encounters is the count of one or more group members coming within six feet of a different group.

[2] Groups is the number of groups observed.

[3] People is the number of people observed.

[4] Percent masked is percentage of people in a group observed wearing masks.

The average number of people observed during the one-minute interval was 11.93 and ranged from 1 to 26. The distrubution of people is shown in Fig 7.

Mask wearing showed a split among visitors in that 61 percent of observed individuals were wearing masks. Within groups there was a distinct split with nearly everyone wearing a mask or no one wearing a mask. In Fig 8 we categorize groups by the proportion of members wearing masks. Thirty-four percent of groups had no members wearing a mask while 55 percent of groups had everyone wearing a mask. Eleven percent of groups had some, but not everyone, wearing a mask. In the analysis, we include the percent of people observed wearing a mask in the group.

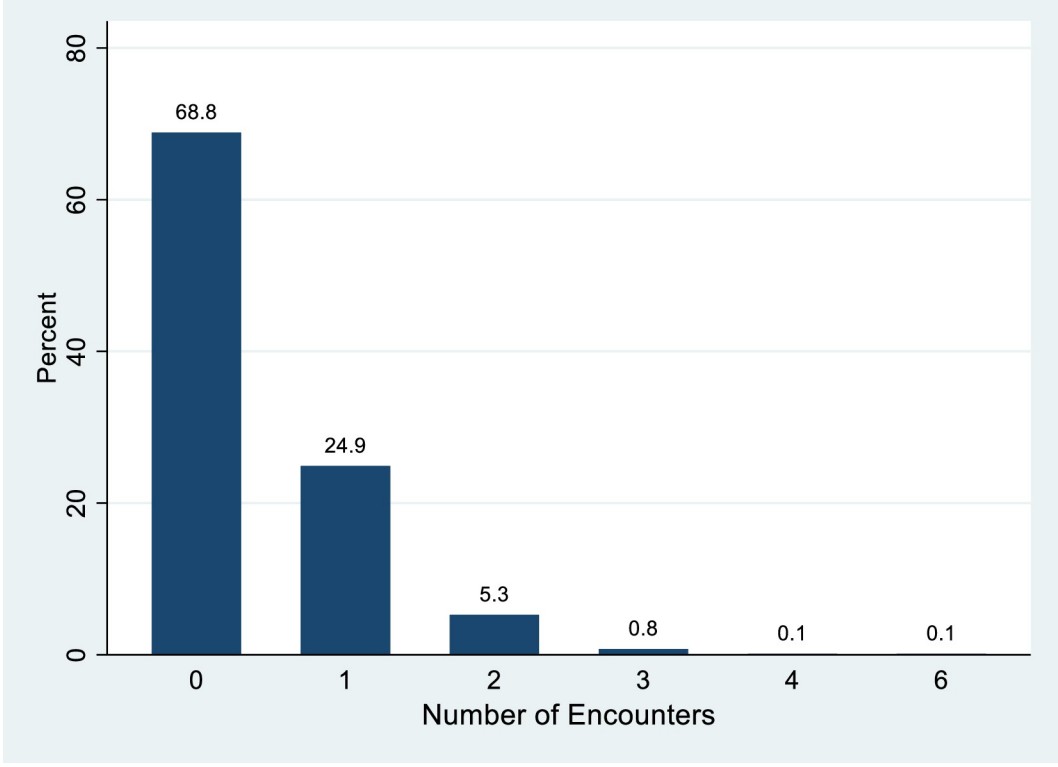

**Fig 4. Percent distribution of number of encounters.**

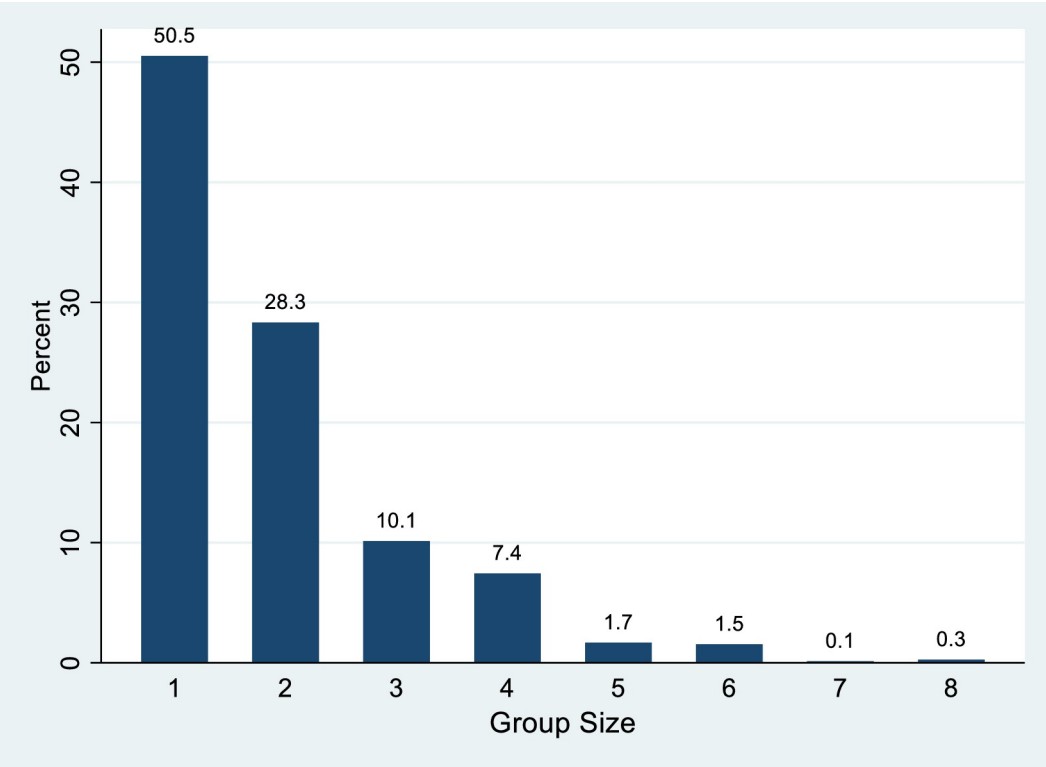

**Fig 5. Percentage distribution of group size.**

### Explaining the probability of encounters

We employ a count model for encounters. Since our count does not display over dispersion (mean encounters = 0.39, variance = .44) we apply a Poisson model with robust standard errors. The results from using group size, number of groups, the number of people present, and the percentage of the group wearing masks appear in Table 2 as Model 1. The Likelihood Ratio test provides convincing evidence to reject the hypothesis that the covariate coefficients are jointly zero ($\chi^2$ = 53.93, $df$ = 5, $p$<0.001). The estimated model coefficients have the expected signs with both group size and the number of groups showing strong evidence that their effect is not zero. Model 2 drops the number of people and the percentage of people in a group wearing a mask.

Model 2 does a good job of modeling the probabilities of encounters as shown in Table 3 where the actual and predicted probabilities from the sample show a close fit. From a practical perspective it is more useful to examine how varying group size and the number of groups impacts the probability of an encounter (see Fig 9 below).

Fig 9 uses model 2 to show how the probability of an encounter changes over group size and the number of groups present. With four groups present, the probability of one or more encounters increases from 19 percent to 40 percent as we go from a group size of 1 to 8 people. If eight groups are present, the probability of one or more encounters increases from 34 percent to 64 percent as the group size increases from 1 to 8. Managing COVID-19 exposure is about individual risk management. The encounters we observed were brief and outdoors which mitigates the risk. However, these were observations lasting only one minute, individuals have the potential for many interactions over the visit and ARCH sees over 1.5 million visitors a year, so small changes in probabilities have the potential for large effects.

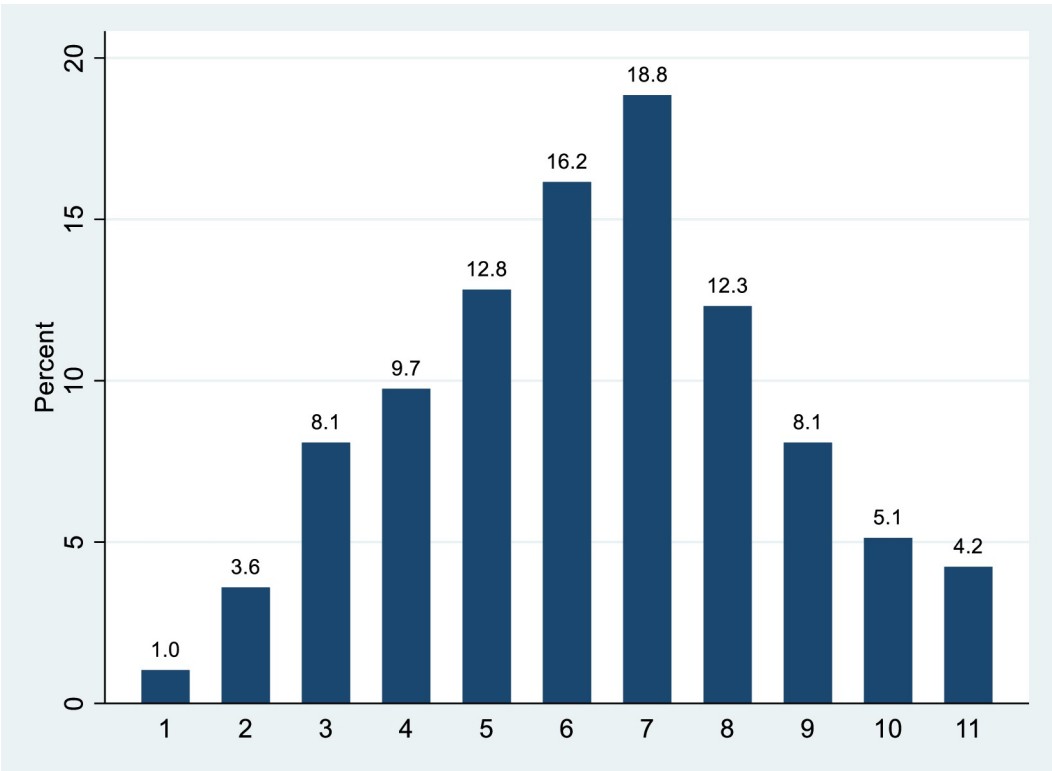

**Fig 6. Percentage distribution of groups at one time.**

## Discussion

Ultimately, the purpose of this research was to determine if visitors are able to socially distance in a crowded national park. Using observations near the visitor center at ARCH, we found evidence through our modeling that most visitors (68.8%) are able to avoid encounters with other groups. However, increasing the number of groups and the number of people in groups both drove increased risk of encounters. In sum, the large majority of visitors are able to navigate these crowded spaces in a way that reduces their risk of encounters. Part of the reason for this is likely due to the fact that so many people are traveling as a single-person group. For example, in our sample over 50% of the visitors were in a single person group. Research in the summer of 2016, demonstrated that only 7% of the visitation to ARCH were in a group size of one [13]. It is possible more people are traveling alone, however it is also conceivable that visitors are either taking turns to approach the facility or only one member of the group is choosing to visit the visitor center, enabling the rest of the party to remain in the parking lot.

Part of this work was also to provide basic descriptive information about visitor compliance with CDC guidelines. We found over 60% of the visitors observed chose wore a face covering. This appears to conform with the health department's guidance asking people to wear face coverings outdoors "when social distancing of 6 feet [1.83 meters] is not possible, reasonable, or prudent." However, we also found that face coverings were not predictive of encounters with other groups. In a real sense, this means that groups not wearing face coverings are no more or less likely to have encounters with other groups. Thus, the roughly 40% of visitor who do not wear masks do pose some risk of disease transmission through <6 foot (1.83 meter) encounters with other groups.

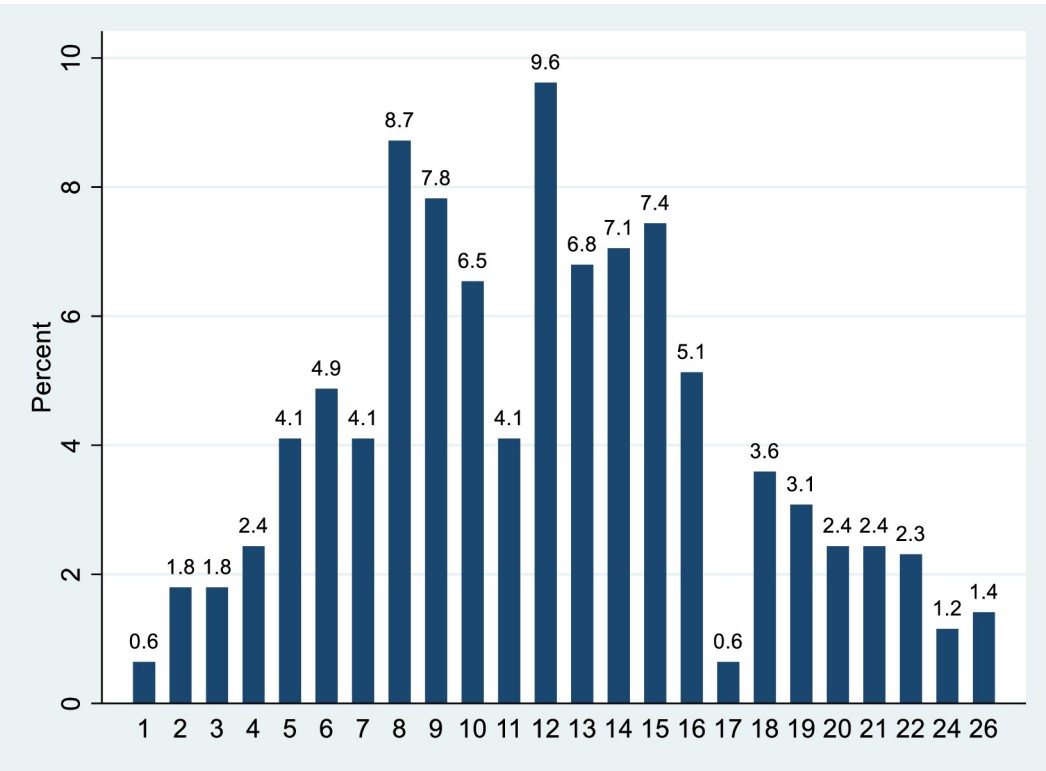

**Fig 7. Percentage distribution of people at one time.**

A follow-up line of inquiry to whether visitors are complying with CDC guidelines is whether there is a cumulative impact of that behavior on the safety conditions within the park. Indeed, while the average number of groups present during our sampling periods was over six, 69% of the visitor groups were able to move through the foyer of the visitor center and back without having an encounter within 6 feet (1.83 meters) of another group. Our sample enabled us to watch visitor behavior when a wide range of groups were present and across different group sizes. The risk of an encounter with another group does increase as the number and size of groups present grows. While overall risks of an encounter are relatively low, a person travelling in a group of one can cut their risk of an encounter by half under most conditions relative to a group size of four.

How far can we extend these results to broader park use? Our study site provided ample opportunity for groups to skirt one another which may not be as possible in areas such as narrow trails. Visitors may also interpret what is reasonable or prudent differently in areas of the park that are less developed, and thus, be less likely to use risk-reducing behaviors. Further research will need to examine behavior on trails and how it changes as the trail sizes decrease.

## Conclusion

Visitors are generally finding ways to avoid close encounters in a concentrated visitor use area of a busy national park. The communication measures taken by park managers nationwide seem to be having an effect and it is encouraging that our parks may remain a relatively safe alternative for people who are seeking constructive ways to escape indoor quarantines and improve both their mental and physical health. The ability for parks to provide these safe havens, however, is not infinite. As use levels or groups sizes increase, the potential for close

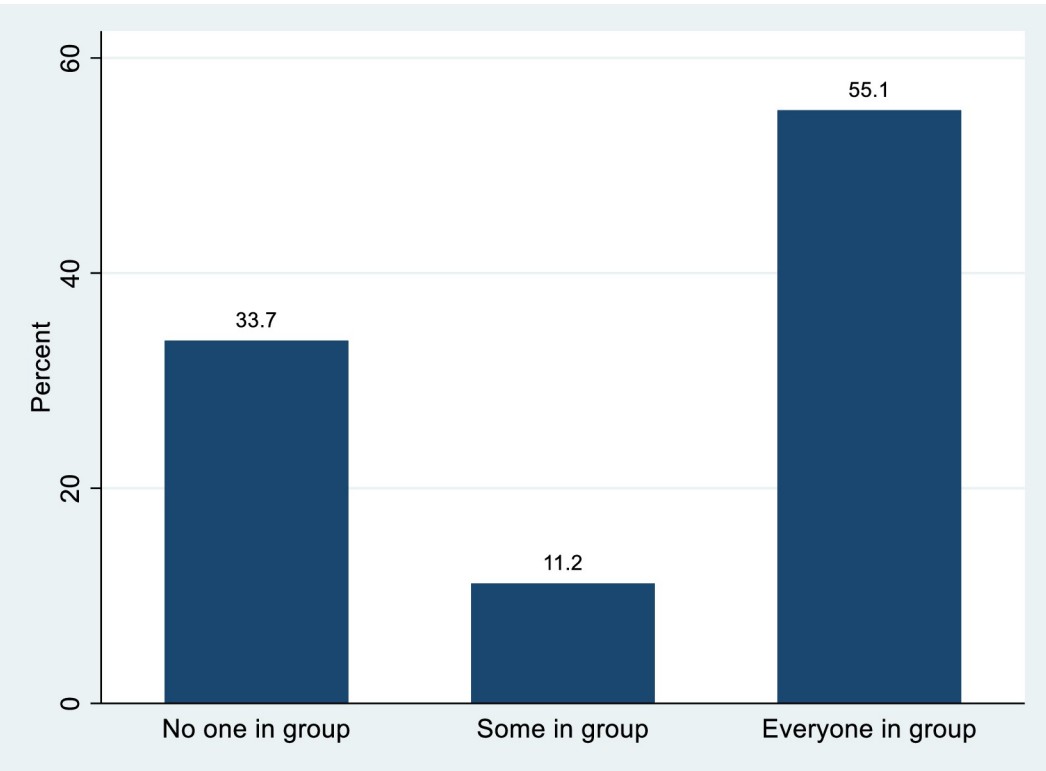

**Fig 8. Percentage of distribution of groups by mask wearing.**

**Table 2. Estimation results.**

| | Model 1 | Model 2 |
|---|---|---|
| | Poisson | Poisson |
| | Encounters | Encounters |
| Group Size | 0.103* | 0.124** |
| | (0.045) | (0.041) |
| Groups | 0.128** | 0.169** |
| | (0.046) | (0.028) |
| People | 0.021 | |
| | (0.018) | |
| Mask pct | 0.000 | |
| | (0.001) | |
| Constant | -2.312** | -2.335** |
| | (0.232) | (0.215) |
| N | 780 | 780 |
| Pseudo R-square | 0.042 | 0.041 |
| Chi-square | 53.93** | 49.90** |
| p-value | <0.001 | <0.001 |
| AIC | 1234.485 | 1231.735 |

Note

* $p<0.05$

** $p<0.01$. Robust standard errors in parentheses.

**Table 3. Actual and predicted probabilities of encounters.**

| Encounters | 0 | 1 | 2 | 3 | 4 | 5 | 6 |
|---|---|---|---|---|---|---|---|
| Actual | 0.688 | 0.249 | 0.053 | 0.008 | 0.001 | 0 | 0.001 |
| Predicted | 0.686 | 0.250 | 0.053 | 0.009 | 0.001 | 0 | 0 |

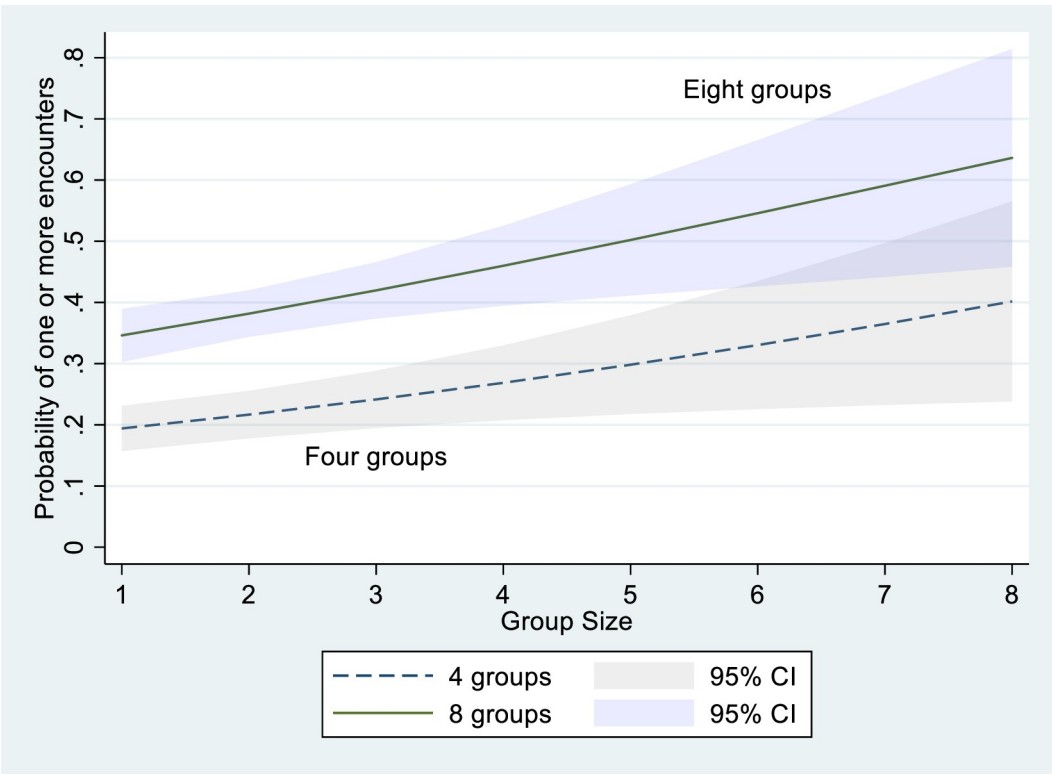

**Fig 9. Probability of an encounter over group size and the number of groups present.**

encounters also increases. We recommend that park managers encourage visitors to travel in small groups whenever possible, but especially in congested areas.

## Acknowledgments

Our research team would like to thank the ARCH staff for assisting us during the research process. Specifically, we would like to thank Amy Tendick for her help. Additionally, we would like to thank Cody Dems for his help with data collection and Wyatt Traughber for recording the photographs.

## Author Contributions

**Conceptualization:** Zachary D. Miller, Wayne Freimund, Douglas Dalenberg.

**Data curation:** Zachary D. Miller, Wayne Freimund, Douglas Dalenberg, Madison Vega.

**Formal analysis:** Zachary D. Miller, Wayne Freimund, Douglas Dalenberg, Madison Vega.

**Investigation:** Zachary D. Miller, Wayne Freimund, Douglas Dalenberg.

**Methodology:** Zachary D. Miller, Wayne Freimund, Douglas Dalenberg.

**Project administration:** Zachary D. Miller, Wayne Freimund.

**Resources:** Zachary D. Miller, Wayne Freimund.

**Software:** Wayne Freimund.

**Supervision:** Zachary D. Miller, Wayne Freimund.

**Visualization:** Zachary D. Miller, Douglas Dalenberg.

**Writing – original draft:** Zachary D. Miller, Wayne Freimund, Douglas Dalenberg, Madison Vega.

**Writing – review & editing:** Zachary D. Miller, Wayne Freimund, Douglas Dalenberg, Madison Vega.

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
