## [Decision Letter · Decision Letter 0]

15 Jan 2021

PONE-D-20-38622

Observing COVID-19 related behaviors in a high visitor use area of Arches National Park

PLOS ONE

Dear Dr. Miller,

Thank you for submitting your manuscript to PLOS ONE. After careful consideration, we feel that it has merit but does not fully meet PLOS ONE’s publication criteria as it currently stands. Therefore, we invite you to submit a revised version of the manuscript that addresses the points raised during the review process.

Only one reviewer accepted our invitation to review.  I read your paper. I enjoyed reading it as well. I believe your methodology is reproducible in other similar scenarios as well.

We look forward to receiving your revised manuscript.

Kind regards,

Itamar Ashkenazi

Academic Editor

PLOS ONE

Journal Requirements:

3. We note that Figure 1 includes an image of a participant in the study. 

Reviewers' comments:

Reviewer's Responses to Questions

**Comments to the Author**

1. Is the manuscript technically sound, and do the data support the conclusions?

Reviewer #1: Partly

2. Has the statistical analysis been performed appropriately and rigorously? 

Reviewer #1: Yes

3. Have the authors made all data underlying the findings in their manuscript fully available?

Reviewer #1: Yes

4. Is the manuscript presented in an intelligible fashion and written in standard English?

Reviewer #1: Yes

5. Review Comments to the Author

Reviewer #1: Dear Authors,

In the paper you tackle an interesting and very current problem, which might seem too obvious to explore at a first glance, yet the findings have the potential for direct positive change. The paper is well written and easily comprehensible – I enjoyed reading it. Consequently, I only note some minor suggestions for improvements.

I whish you all the best in the New Year and you future work

Detailed list of suggestions:

- Please provide SI units throughout the paper.

- Line 69: I suggest adding ”entire” before April (to distinguish that is is not part of March and April)

- Paragraph starting with Line 83: please provide some references for CDC guidelines and the statements about ”behaviour” of NPS

- Sentence starting in line 100: It makes me wonder whether the Authors are in any capacity connected to the observed site (not necessarily in sense of conflict of interest, more so to gain insight into why you chose this particular site)

- Sampling details (line 125 onward): Please discuss how the chosen dates are representative of the general (pre-COVID) visitation patterns, and how representative it is to the camera downtime (or do the numbers of visitors drop so significantly that it does not warrant observation?)

- Recording visitor behaviors (line 140 onward): It is not clear whether there was a single coder for the videos. How did you control for the quality of coding?

- Line 152: I wonder whether there was an instance where a person was facing away from the camera and it was thus impossible to determine whether they are wearing a mask. How did you code the data in such case (if applicable)?

- Line 155: Please add a brief discussion on how reliable is the use of reference points to determine distance and/or how it might impact the results if there could be such impact.

- Line 211: I suggest removing the word ”only” as we should avoid evaluating the results before discussion.

- Explaining the probability of encounters (line 217 onward): Please elaborate/provide more detail (I feel this part is significant, as you go beyond ”counting” and provide novel insights). What is the practical significance of the model results (e.g., pseudo r2)?

- Line 221: Did you mean Table 2?

- Line 240: This statement could also be provided somehow in the introduction (as a research aim)

- Please elaborate the discussion by adding some critical evaluation of the results and their use. You state in the first sentence of the Discussion that you wanted to determine whether visitors comply with CDC guidelines, but then you only provide the percentage of visitors wearing mask without discussing the actual finding. The other percentages also seem moderate, and some statements in the first paragraph of page 12 seem a bit ambitious. You provide some recommendations and discuss the possibility of generalization of the findings, but I was missing something beyond describing the results. Were your research questions answered? How are the results useful for this and/or other parks?

- Sentence starting in line 248: How could you determine that from your observations? Could we also stipulate that the person came alone (and that there is a rise of one-person groups now)?

- Line 254: How do you interpret this percentage? Is that enough/a lot/not enough? Please elaborate your discussion.

- Line 271: Seems a bit ambitious.

- Line 273: Consider replacing ”notoriously” (this was not the impression I got from the paper, and I personally never heard of the park before so all the information I have about the park were provided by you).

- Sentence starting in line 274: Based on your observations, you cannot make draw causal conclusions. I suggest rewording this part of the paragraph.

- Line 278: How do your recommendations differ from the current recommendations to the visitors? Are they viable in practice?

- Figure 2: Please provide time-span for the plotted data

- Figure 8: Consider using the full word ”group”

- Figure 9: Please provide the label for y-axis

6. PLOS authors have the option to publish the peer review history of their article (what does this mean?). If published, this will include your full peer review and any attached files.

Reviewer #1: **Yes: **Žan Lep

---

## [Author Response · Author response to Decision Letter 0]

1 Feb 2021

We have provided a detailed response to reviewers table in the attachments

---

## [Editor Report · Decision Letter 1]

5 Feb 2021

Observing COVID-19 related behaviors in a high visitor use area of Arches National Park

PONE-D-20-38622R1

Dear Dr. Miller,

We’re pleased to inform you that your manuscript has been judged scientifically suitable for publication and will be formally accepted for publication once it meets all outstanding technical requirements.

Kind regards,

Itamar Ashkenazi

Academic Editor

PLOS ONE
---

## [Editor Report · Acceptance letter]

8 Feb 2021

PONE-D-20-38622R1 

Observing COVID-19 related behaviors in a high visitor use area of Arches National Park 

Dear Dr. Miller:

I'm pleased to inform you that your manuscript has been deemed suitable for publication in PLOS ONE. Congratulations! Your manuscript is now with our production department. 

Kind regards, 

on behalf of

Dr. Itamar Ashkenazi 

Academic Editor

PLOS ONE